# Effects of Thermal Manipulation on mRNA Regulation of Response Genes Regarding Improvement of Thermotolerance Adaptation in Chickens during Embryogenesis

**DOI:** 10.3390/ani12233354

**Published:** 2022-11-29

**Authors:** Suriya Kumari Ramiah, Krishnan Nair Balakrishnan, Yashini Subramaniam, Oluwaseun Serah Iyasere, Zulkifli Idrus

**Affiliations:** 1Laboratory of Sustainable Animal Production and Biodiversity, Institute of Tropical Agriculture and Food Security, Universiti Putra Malaysia (UPM), Serdang 43400, Malaysia; 2Department of Animal Physiology, Federal University of Agriculture, Abeokuta P.M.B 2240, Nigeria; 3Department of Animal Science, Faculty of Agriculture, Universiti Putra Malaysia (UPM), Serdang 43400, Malaysia

**Keywords:** heat stress, heat shock proteins, thermal manipulation, embryogenesis, antioxidant, immune genes, chicken, thermotolerance adaptation

## Abstract

**Simple Summary:**

Heat stress is a threat to poultry, affecting both animal welfare and productivity. Temperature manipulation during embryogenesis is considered one of the mitigation strategies with which to reduce the negative impact of high ambient temperature. However, the mRNA regulation underlying genes that are responsible for thermotolerance acquisition during embryogenesis remain unclear. The pattern of gene expression, focusing mainly on heat shock proteins, antioxidants, and immunological genes, of chickens that have undergone thermal manipulation during their embryonic stages was emphasized in this review. Considering the importance of animal welfare, temperature manipulation during embryogenesis indicated the enhanced adaptability of chickens towards encountering heat stress at later stages in life.

**Abstract:**

The phenomenon of increasing heat stress (HS) among animals is of particular significance when it is seen in economically significant industries, such as poultry. Due to the identification of the physiological, molecular, and genetic roots of HS responses in chickens, a substantial number of studies have focused on reducing the effects of HS in poultry through environmental management, dietary manipulation, and genetic alterations. Temperature manipulation (TM) during embryogenesis has been claimed to increase the thermal tolerance and well-being of chickens without affecting their capacity for future growth. There has been little investigation into the vulnerability of the epigenome involving TM during embryogenesis, although the cellular pathways activated by HS have been explored in chickens. Epigenetic changes caused by prenatal TM enhance postnatal temperature adaption and produce physiological memory. This work offers a thorough analysis that explains the cumulative impact of HS response genes, such as genes related to heat shock proteins, antioxidants, and immunological genes, which may aid in the enhanced adaptability of chickens that have undergone thermal manipulation during their embryonic stages.

## 1. Introduction

Animal husbandry has faced significant difficulties recently as a result of the rising global temperatures and frequent heat waves brought on by global warming. The poultry industry faces a very serious problem called heat stress (HS), especially in some tropical regions of Asia [1]. When an animal’s metabolism and heat absorption from the environment outweigh its ability to escape the heat through radiation, convection, evaporation, and conduction, the result is heat stress (HS) [2]. Newly hatched chickens are poikilotherms and, as such, cannot control their body temperature during the post-hatching and early developmental phases [3]. Chickens can withstand only a narrow temperature range, and their thermo-neutral zone is between 18 °C and 21 °C [4]. When their body temperature rises due to environmental or other metabolic factors, they cannot disperse enough body heat to maintain thermal equilibrium [5]. Consequently, their growth performance is affected, and they become highly susceptible to diseases and mortalities, which thereby affects their productivity [6,7]. Therefore, precautions such as the management of feeding, lighting, ventilation, litter, nutrition, and housing systems to mitigate the negative effects of HS on chicken performance are important [8]. Alternative approaches include early-age food restriction and allowing broilers to undergo a period of HS [9]. In addition, it has been suggested that thermal manipulation (TM) during embryogenesis improves thermotolerance and poultry welfare without impairing their future growth performance [10,11,12].

The manipulation of temperature during incubation has resulted in an impact on embryonic development and hatchability in poultry. The modification of the incubation temperature of avian eggs during embryogenesis depends on the frequency, duration, amplitude of temperature, and embryo age with respect to the timing of TM. Studies from several researchers have shown conflicting findings after an alteration of the amount of TM exposure. In addition, most of these studies report only the effects of temperature manipulation on chick weight at hatch and on hatchability, body weight, heat production, and the hormonal levels of post-hatch chicks, as shown in Table 1.

**Table 1 animals-12-03354-t001:** The effects of temperature manipulation on chick weights at hatch and on hatchability, body weight, heat production, and the hormonal levels of post-hatch chicks.

No.	Thermal Manipulation (TM) Temperature	Thermal Manipulation Age/Embryonic Day (E)	Results	References
1.	39.5 °C for 12 h (Intermittent) and 24 h/day Continuous)	E7 to E16	Plasma corticosterone levels at hatch were significantly higher in the TM-treated chicks No difference in hatchability between the control and TM group The development of the thyroid and adrenal axis lowered their functional set points during TM	[13]
2.	39.5 °C for 12 h (Intermittent) and 24 h/day Continuous)	E7 to E16.	Continuous TM enhanced thermotolerance on heat stress at days 2 and 10 of age	[14]
3.	39.5 °C	3 h on E8 to E10 (early: EA) 3 h on E16 to E18 (late: LA)	The EA and LA chicks exhibited significantly low body temperature at hatching Late treatment enhanced breast muscle growth Improved long-term thermotolerance in chickens	[15]
4.	39.5 °C	3, 6, 12 or 24 h per day from E16 to E18	Recommended that 3 h of TM per day during E16 to E18 improved thermotolerance acquisition. Higher hatchability and growth	[16]
5.	38.8 ℃	6 h on E 10 and 14.	Increased hatch weight and 42 d body weight Low plasma triacylglycerol at 42 d Increased insulin-like growth factor-I (IGF-1) gene expression	[17]
6.	39.5 °C	12 h/day on E 7 to 16	Lower body temperature at hatching Changes in plasma thyroid hormone concentrations. Breast muscle yield was enhanced	[18]
7.	38.5 °C	3 h/day on E16,17 and 18	Plasma concentration of thyroid hormones (T4 and T3) was lower No effect of the treatment on the post-hatch plasma corticosterone concentration No effect on the hatchability	[19]
8.	34.6 °C (low) 37.6 °C (control), 40.6 °C (high) for	4 h/embryonic day from E16–E18,	Embryos in the high-temperature treatment group shifted toward a more anaerobic metabolism ✓ lower partial pressure of O_2_ and a higher partial pressure of CO_2_ in the air-related cells ✓ lower blood pH ✓ higher lactic acid production Embryos in low-temperature treatment group shifted toward lower metabolism. ✓ higher plasma triglyceride level; higher liver glycogen level; higher relative yolk weight	[20]
9.	39.0 °C for	2 h on E13–E17	Downregulation of T4 Elevated blood pH Increased O_2_ demand Upregulation of corticosterone synthesis	[21]
10.	38.2 to 38.4 °C for 24 h daily (chronic warm incubated) and 38.2 to 38.4 °C (warmth-stimulated).	2 h daily on E 18 until hatch	Female chickens adapted well to warm conditions and performed better (feed intake, body weight gain, and body weight) during the final growing period The daily feed intake for male chickens incubated under chronic temperature was negatively affected during the final growing period No effect on the hatchability and chick quality.	[22]
11.	39.6 °C	6 h/day on E 0–8 (early) 6 h/day E10 to 18 (late)	Applications of TM in the early and late periods of incubation: ✓ Decreased hatchability rate ✓ No adverse effects on embryo morphology, development balance, and chick weight High temperature in late embryonic days reduced chick quality.	[23]
12.	38.5 °C and 40 °C	6 h on E16, 9 h on E17 12 h on E18	Hatchbility not affected Body weights of thermally manipulated chickens were higher compared with controls Significant reduction in the plasma level of corticosterone in TM groups	[24]
13.	38.4 °C	6 h on E 19 3 h on E19 to 20	Higher body weight and body weight gain in broiler chickens Better FCR compared to control Low feed intake Improved FCR Male broilers obtained higher body weight with embryos exposure to higher temperature for 3 h in days 19 and 20 Post-hatch performance of broiler chickens in TM of broiler embryos during late embryogenesis period were improved	[25]
14.	39.5 °C 39.5 °C	Group 1: 3 h on E4–6 Group 2: 90 min and then returned to 37.5 °C for 60 min and lastly the temperature was raised again for 39.5 °C for 90 min on E4–6	Low hatchability rate in thermally manipulated chickens Body surface temperature and cloacal temperature increased in thermally manipulated chickens Total weight gain and daily weight gain lower in group 1 Triiodothyronine (T3) hormone concentration and globulin levels were low in thermally manipulated chickens	[26]
15.	39 °C	2, 3 and 4 h per day on E16–18	No significant effect on hatchability traits and embryonic mortality percentages No significant differences in body temperature in all TM-treated groups Decreases in plasma total cholesterol and LDL cholesterol 4 h/day TM at E16 to 18 days improve thermo tolerance acquisition, improved chick quality and productivity and low embryonic mortality	[27]
16.	39.5 °C	3 h/day on E11–16	TM treated group has no effcet on hatchability and embryonic mortality No significant differences in the post-hatch performance and quality of the chicks Eggshell temperature and serum concentrations of corticosterone were found to be higher in the TM group	[28]

TM during embryogenesis changes gene expression by altering DNA methylation, which improves future thermotolerance adaptation [29,30]. Prenatal TM has been reported to cause epigenetic changes that result in physiological memory, which improves thermotolerance adaptation in adult life [31,32]. The expressions of mRNA levels targeting different heat shock genes at different time points were primarily investigated by Al-Zhgoul et al. [33,34] as a further confirmation for the acquisition of thermotolerance. Exploration at the molecular level regarding the continuous exposure of post-hatch birds to HS have extensively conducted [35,36]. Among these studies, the expression of heat shock protein (Hsp)-related genes is the most prominent and extensively studied [37,38,39]. In addition to Hsp, a range of gene expressions of different cytokines are triggered by HS in broiler chickens [40]. HS significantly affects oxidative stress, as high reactive oxygen species (ROS) levels could harm broilers by enhancing lipid peroxidation [41]. Many studies have discussed the messenger RNA (mRNA) expression levels of Hsp, cytokines, and antioxidant enzymes in heat-stressed birds alone; however, there are few studies regarding TM during embryogenesis.

The gene regulation of Hsp, antioxidants, and immunity expression in thermally manipulated broiler chickens has not been fully studied; therefore, this review aimed to elucidate the cumulative effect of HS response genes, including Hsp, antioxidant, and immune genes, which may contribute to the improved adaptability of chickens that are thermally manipulated in their embryonic stages.

## 2. Effect of Thermal Manipulation on the Regulation of Heat Shock Factor (HSF)

The chicken embryo responds to the environmental temperature around day 15 of incubation when the hypothalamus–pituitary–thyroid (HPT) and hypothalamus–adrenocortical axes (HPA) control the thermoregulatory mechanism and manage stress responses, respectively [42]. An increased incubation temperature during the development of the HPT and HPA axes could have long-lasting effects on the body function and behavior in chicks post-hatch [17].

Four transcription factors, known as Hsfs, namely, Hsf1, Hsf2, Hsf3, and Hsf4, regulate and activate Hsp-related genes [43]. Hsf1 and 3 are the main Hsfs in avian species responsible for activating the heat shock element (HSE), an upstream promoter region in the heat shock gene when chickens are subjected to stress conditions [44]. It was discovered that changes in Hsfs’ mRNA expressions were correlated with changes in Hsps’ mRNA expressions [45]. In addition, Hsf activation triggers the expression of Hsp genes in response to elevated temperatures and long-term exposure [46]. Al-Zghoul [43] revealed that low Hsf3 gene expression correlated with high Hsp70 mRNA levels. The thermal challenge from embryonic days (ED) 12 to 18 significantly increased Hsf-1 levels in the heart and brain tissues of broiler chicks on post-hatch days compared with that in the controls [47]. Notably, similar changes in the mRNA expression of Hsf1 and 2 were observed in the muscles of thermally manipulated chicks at ED 12 and 18 and on post-hatch days [48]. This suggests the spatial expression of Hsf genes between different tissues [49]. Notably, the activation of heat-inducible Hsf 1 and 3 has been demonstrated to require a specific temperature threshold [50]. However, whether there is a temperature threshold for activating gene transcription in Hsfs remains unknown.

### 2.1. Effects of Thermal Manipulation on the Regulation of Heat Shock Proteins (Hsp) 60, 70, 90, and 108 and Small Heat Shock Proteins (sHsps) on the Muscle, Heart, Brain, and Gastrointestinal of Chickens

Hsps are highly conserved proteins that are rapidly generated in organismal tissues in response to physical, chemical, or biological stimuli, such as heat exposure in chickens [7,51]. Hsps principally functions are facilitating protein structure and fold formation and promoting cell survival [49,52]. The classification of Hsps based on their high or low molecular weight varies by their ATP requirements for proper functionality. Therefore, Hsps with high and low molecular weights are termed as “ATP-dependent” or “ATP-independent”, respectively [53].

#### 2.1.1. Hsp 70

Hsp70 is possibly most notably associated with thermal tolerance [34]. It is involved in the chaperon system in protein folding, homeostasis, the prevention of protein aggregation, and cell survival promotion in stressful environments [36]. Different TM protocols resulted in a significant increase in Hsp70 mRNA expression in the pectoral and thigh muscles [33,34], and the same was associated with enhanced thermotolerance, as previously reported by Al-Aqil and Zulkifli [9,54]. Ali et al. [49] suggested that TM enhanced the expression of Hsp70 mRNA [33,47,48,55,56] in chickens. In addition, Hsp70 nucleotide and amino acid sequences are more constant in thermally manipulated chicks during their embryonic and post-hatch lives [49].

One school of thought suggested that chicks under embryonic thermal conditions could better adapt to external temperature stress resulting from decreased Hsp gene expression [5]. Vinoth et al. [5] investigated the effects of TM on the Hsp genes of two breeds, naked neck (NN) and Punjab Broiler-2 (PB2), during EDs 15 to 17; resultantly, the heat-exposed birds from both breeds that had undergone thermal conditioning exhibited lower levels of Hsp genes (Hsp70, 27, 60, and 90 alpha; 90 beta; and ubiquitin). The brain tissue of the thermally altered chicks had reduced levels of Hsp70 mRNA expression during a subsequent post-hatch thermal challenge, indicating that small changes in brain temperature are insufficient with respect to causing an accumulation of Hsp70 in this tissue [47]. These findings reinforce the notion that some species exhibit mechanisms that do not allow for a change in brain temperature, which is maintained at a temperature 2–3 °C lower than the rectal temperature [57]. In support of the aforementioned results, this indicates that the differential expression of Hsp70 mRNA is temperature- and tissue-dependent in chickens.

Vinoth et al. [12] studied the effect of thermal conditioning during embryogenesis and a thermal challenge at 42 days of age on the Hsp gene. Interestingly, it was observed that epigenetic changes that affect the DNA methylation of the Hsp70 promoter could cause shifting expression dynamics, hence modifying the expression of proteins during the post-hatch life of broiler chickens [12]. In contrast to a previous study by Al-Zghoul [43], Shanmugasundaram et al. [58] demonstrated that greater basal Hsp70 expressions in a TM group than in the control group indicate that TM has both short- and long-term impacts on gene expression in ducks. This might indicate that thermal manipulation may have differences between species.

#### 2.1.2. Hsp 60

Hsp60 is a node in intracellular molecular networks and is mainly localized in the mitochondria of eukaryotic cells [59]. Furthermore, the punctiform dispersal of Hsp60 in the mitochondria of heat-stressed cells has been reported [16]. It was revealed that TM between EDs 12 and 18 could increase Hsp60 mRNA expression, which is higher in the heart and muscle than in the brain [43,47]. This indicated that mRNAs of Hsp60 may be transported into the mitochondria under HS and tissue specificity [16,43]. Therefore, Hsp60 might be an additional biomarker in response to heat.

#### 2.1.3. Hsp 90 and Hsp 108

Hsp90 is essential for maintaining protein homeostasis and is associated with proteins throughout the late developmental stages in broiler chickens [46,47]. Hsp108 is constitutively produced in numerous chicken tissues and induced by HS in primary cell cultures and chicken oviducts [60]. Significant increases in the basal expressions of Hsp108 and 90 during embryogenesis and post-hatch in broiler chickens were observed after various TM methods were applied [5,45,56]. The epigenetic alterations caused by TM may have altered these genes by improving thermotolerance and tissue stability acquisition during hyperthermia [45]. How Hsp108 contributes to thermotolerance in chickens is unclear. Limited information is available on the role of mRNAs of other Hsps such as Hsp108 or Hsp27 and the underlying mechanism by which they promote stress tolerance in chicks post-hatch.

#### 2.1.4. Small Heat Shock Proteins (sHSPs)

Most studies have focused on large Hsps such as Hsp90, 70, and 60 with respect to TM during embryogenesis; therefore, the significance of sHsps in poultry thermotolerance is poorly understood. sHsps are the least-conserved Hsps, with low-molecular weight proteins (15–43 kDa). Basaki et al. [61] suggested that the TM of broiler chicks during embryogenesis increases sHsp expression in the brain. Different sHsp isoforms have been reported in animal studies, all of which express and change the transcriptional levels in skeletal muscle to protect cells under stress environments [46,62]. However, limited information is available regarding the expression of sHsp genes in thermally manipulated chicks at hatching. The sHsps gene is a potential candidate stress-related biomarker for thermally manipulated broiler chickens.

The chicken intestinal tract is considerably prone to HS, which causes intestinal integrity impairment, inflammation, and marked alterations in the gut microbiota [54,63]. Specific alterations have been reported to disrupt Hsps expression between intestinal segments (duodenum, ileum, and jejunum) of poultry subjected to HS conditions [64,65]. Khaleel et al. [66] and Al-Zghoul and Saleh et al. [55] reported that thermally manipulated broiler chickens showed an altered post-hatch response to chronic HS in the jejunal mucosae by reducing the expression of Hsp70 and Hsf1 and 3. Similarly, embryonic TM from day 11 of incubation until hatching improved intestinal integrity and reduced Hsp70 expressions [67]. This indicates that TM might have a long-lasting impact on these genes, which were associated with an improved small intestinal response to post-hatch thermal exposure compared to control chicks with higher levels of Hsp mRNA.

In the jejunal digesta of broiler chickens under heat stress, there was a strong positive association between HSP70 expression and digestive enzyme activity (amylase, lipase, and trypsin activity) [68] This suggests that increasing intestinal digestive enzyme activity by HSP70 overexpression may improve intestinal digestion and absorption in broilers subjected to acute heat stress. According to Al-Zghoul et al. [34], the TM of broiler chickens has a long-term impact on their nutritional digestion and absorption capacity during chronic heat stress. However, further research into the relationship between intestinal digestion, absorption, and heat-stress-induced transcription factors in thermally manipulated chicks is intriguing. Therefore, an investigation into the mechanisms whereby the incubation temperature affects the intestinal integrity of thermally manipulated chicks needs to be carried out comprehensively.

### 2.2. Effect of Thermal Manipulation on the Regulation of mRNA Levels of Cytokines

Besides Hsps, a range of different gene expressions of cytokines were found to be triggered by heat stress in broiler chickens. These cytokines are responsible for cell-signaling polypeptides’ ability to regulate the immune response in different circumstances and, importantly, to repair and regenerate the damaged tissue resulting from heat stroke [69,70]. Different types of cytokines are classified as interferons (IFN), interleukins (IL), chemokines, transforming growth factor (TGF) lymphokines, and tumor necrosis factor (TNF) based on their function, site of secretion, or the cells that they act upon [71]. Many studies have discussed the roles of cytokines in heat-stressed birds alone; however, fewer studies involve TM during embryogenesis. An initial study on TM during embryogenesis and the effects on cytokine expression was carried out by Al-Zghoul et al. [47]. The authors demonstrated an increased expression of IL (interleukin)-6 and IL-1β among thermally manipulated chickens compared to the controls in response to AHS (acute heat stress). Notably, a similar pattern of expression was observed for Cobb and Hubbard breed chickens. Besides HS, both genes were also found to be upregulated during artificially induced stress. Therefore, the possible reasons for this observation could be due to the involvement of IL-6 and IL-1β in tissue protective pathways and the activation of an acute phase response [47,70]. In addition, IL-6 was identified to play the role of a heat shock gene during HS, since its transcription is triggered by Hsf 3, and the higher expression is associated with protection in the yellow follicles of chickens [40,72]. Next, other interleukins such as IL-18, IL-17, IL-8, IL-15, and IL-16 were found to be increased in thermally manipulated chicks during AHS compared to controls and it could be concluded that the majority of interleukins increase their expression to mitigate the adverse effects [48]. However, the same study discovered a lower expression of IL-4 in thermally manipulated chicks during AHS and it was expected to correlate with an increased expression of IFN-γ (interferon-gamma). Since the immune system is functionally polarized, either pathway of IFN-γ or IL-4 will facilitate protection [73,74]. A similar uptrend in expression for IL-6 and IL-1β together with TNF-α (Tumour Necrosis Factor-alpha), IL-18, IL-17, IL-8, and IL-16 was encountered in splenic broiler embryos [48]. This correlates with the expression of selected interleukins between thermally manipulated embryos and post-hatch chickens. Next, the same authors focused on the expression of interferons among thermally manipulated chickens in response to AHS. IFN-α (interferon-alpha) expression was found to increase significantly during AHS in both TM and control; however, the expression was much lower in the TM group compared to the control. On the other hand, IFN-β (interferon-beta) expression was observed to be without changes in the TM group during AHS, while a significant reduction was encountered in the control group. The reason for the discrepancies is unclear; however, it is known that IFNs are more responsible for antiviral activity and might be less responsive to HS [75]. The aforementioned discussions focused on TM’s effects on embryogenesis on animals subjected to AHS while Al-Zghoul and Saleh [55] investigated the interleukin expression in thermally manipulated chickens subjected to CHS (chronic heat stress). Interestingly, the thermally manipulated chicks expressed higher mRNA levels of IL-6 and IL-8 after 1 day; however, the expression of IL-1β, TNF-α, and IL-6 were noticed to be lower after 3, 5, and 7 days of CHS compared to the controls. The higher expression of IL-6 after 1 D could be attributed to the increased expression of Hsf 3 on the same day. Again, this strengthens the notion that HSF3 activates the expression of IL-6, which could play a role as a heat-shock gene, as mentioned previously [40]. Besides that, the CHS findings are in contrast with AHS, which has a higher expression of IL-6, IL-1β, and TNF-α. The differences in results might be due to the stress conditions (CHS and AHS) and the type of organs and tissues included in the study (jejunal mucosae for CHS while liver and spleen for AHS). Therefore, further studies are required regarding CHS to more comprehensively understand the expression levels of interleukins in different tissues and organs

### 2.3. Effect of Thermal Manipulation on the Regulation of mRNA Levels of Antioxidant Enzymes

Many studies have documented a higher expression of mRNA levels of antioxidant enzymes SOD (Superoxide dismutase), NADPH oxidase (NOX), glutathione peroxidase 2 (GPX), and catalase as a preventive measure to minimize or halt the oxidative stress in chickens during HS [76]. HS has a significant effect on oxidative stress, as high levels of reactive oxygen species (ROS) could be harmful to broilers via enhancing the lipid peroxidation process. Generally, increased ROS levels are attributed to protein denaturation, antioxidant enzyme alteration, lipid peroxidation, and DNA damage [76]. The most important membrane-bound protein complex known as NOX serves as a major oxygen sensor [77]. NOX’s function is to protect biological systems from severe inflammatory stresses. The main function of NOX is transporting electrons to oxygen molecules from NADPH-producing superoxide (O_2_-) followed by transformation into oxygen molecules (O_2_) or hydrogen peroxide (H_2_O_2_) [78,79]. SOD is the responsible catalyzer for the conversion of (O_2_-). The remaining (H_2_O_2_) will be converted into oxygen molecules (O_2_) and water with the aid of GPX2 and catalase enzymes [80]. Moreover, avian uncoupling protein (AvUcp), which acts as a proton regulator, helps to protect cells from oxidative stress [81]. Once the ROS activity increases due to the upregulation of NOX, then it will cause a down-regulation of AvUcp to manage the oxidative stress. In terms of mRNA expression, antioxidant enzymes, especially NOX4, GPx2, SOD2, and catalase, have been found to be significantly lower in both Cobb and Hubbard TM groups compared to controls after AHS [47]. Similarly, the TM groups had lower mRNA levels of antioxidant enzymes after CHS than the control group [55]. On a cautionary note, significantly increased expression of antioxidant enzymes mRNA levels was noticed for both TM and control groups after CHS, which is different from the AHS results. This might be due to the difference in heating intensity during both experiments. Before CHS (Day 0), there were no significant changes in the antioxidant enzymes observed in both the TM and control groups. However, after HS, it was reported that the increased expression of NOX was directly proportional to the higher production of superoxide (O_2_-), which elevates the host’s oxidative stress. On the other hand, AvUcp expression is seen to reduce during HS, suggesting an indirect relationship with NOX. Therefore, the studies focusing on TM unveiled a reduction in the expression of NOX4 and an upregulation of AvUcp expression, indicating that efforts being were being exerted to alleviate oxidative stress [47]. However, early age TM reported a contradictory reduction in AvUcp expression in broilers; thus, the period of introducing TM could have a significant effect on the antioxidant enzymes’ expression patterns. Next, the catalase and GPX mRNA expressions were noticed to be increased initially and significantly lower after 7 days following CHS in the TM group than in the controls. Concurrently, lower levels of mRNA expressions of catalase and GPX were reported in the TM group during post-hatch AHS in broilers [47]. For SOD, the mRNA expression was reported to be increased for the first 3 days and reduced significantly after 5 D in the TM group following CHS, while the expression remained lowered in the TM group after 1, 3, and 7 h of AHS compared to the control groups [55]. These results are in agreement with Vinoth et al. [5], who reported that lower blood SOD activity resulted in higher serum lipid peroxidation. This might be due to a reduced release of free radicals due to the decrease in energy metabolism in the birds subjected to TM during embryogenesis. In addition, Loyau et al. [82] showed that reduced liver energy metabolism could lower the heat production during TM, which indirectly resulted in minimal oxidative stress in chickens.

### 2.4. Changes in Gene Activity through Epigenetic Alterations Induced by Thermal Manipulation

Reports have shown that epigenetic temperature adaptation in poultry is dependent on the post-hatching conditions; for example, postnatal heat acclimation led to better temperature tolerance in chickens through histone modifications at BDNF, Eif2b5, and HSP70 in the hypothalamus and the methylation of DNA at BDNF [83,84,85,86]. In addition, the action of TM during embryogenesis regulates gene expression, metabolism, physiology, and performance in chickens [87]. Initial TM experiments performed on embryos mainly targeted selective gene expression and the acquisition of heat stress tolerance. Nevertheless, the mechanisms behind these studies are not fully explained and the researchers hypothesized that epigenetic modifications may have occurred [14,38,88,89,90]. On a molecular basis, Loyau et al. [10] performed a transcriptome analysis of the pectoralis major muscle in TM and control chickens subjected to heat and non-heat challenges in 34-day-old broilers. Notably, 759 genes were found to be differentially expressed between the heat-challenged thermally manipulated and non-heat-challenged thermally manipulated chickens. In particular, 28 genes involved in chromatin remodelling and organization and active RNA splicing were identified [10]. This clearly pointed to the epigenetic changes involved in the chromatin landscape that affect the pattern of gene expression in favour of temperature tolerance. Next, a study by Vinoth et al. [12] demonstrated a high methylation level of HSP70 and HSP90 alpha and beta in a heat-stressed TM group compared to heat-stressed control chickens at 42 days of age. The rate of methylation was found to correspond to an inverse relationship with the expression level of mRNA; thus, a higher expression was noticed in the control group compared to the TM group [12]. Accordingly, other studies in chickens observed similar findings, verifying the role of promotor methylation as a repressive epigenetic marker that can alter gene expression [91,92]. On a cautionary note, methylation frequency differs across various strains and tissues of chickens [12,93]. The induction of the epigenetic effect during embryogenesis was reported by David et al. [31] using the genome-wide mapping of two histone markers, H3K4me3 and H3K27me3, in broiler hypothalamus and muscle tissues. This study revealed that the molecular rearrangements on the DNA-packaging histone protein H3 resulted in stress-adaptability later in life [31]. In addition, TM-induced histone changes have a significant impact on the neurodevelopmental, metabolic, and regulatory functions in the hypothalamus. In another study focusing on the methylome, Corbett et al. [30] performed an epigenome-wide association study (EWAS) in cardiac DNA methylation profiles resulting from different eggshell temperatures and CO_2_ incubation levels. The EWAS results revealed 23 significantly associated CpGs, explaining the epigenetic regulation that has an impact on the heart growth rate during hatching by altering the DNA methylation patterns [30].

## 3. Conclusions

The epigenetic regulation of gene expression has become important for poultry species in response to severe climatic change. Our present study helps us understand how thermal conditioning during embryogenesis triggers the production of Hsp and Hsf genes and boosts the expression levels of interleukins and ROS-scavenging proteins, which enhances chickens’ ability to tolerate heat in their postnatal lives. This demonstrates that chickens can keep an adaptive epigenetic memory of the temperature circumstances during the thermal manipulation of embryos. The enhancement of thermotolerance in hens during their postnatal life will be achieved by ascertaining the molecular mechanisms behind TM-induced gene expression. The regulation of epigenetic mechanisms in chickens is currently only partially understood as a result of investigations using specific genes. Only limited information is available on the epigenetic effect of thermally manipulated embryos and it provides a useful framework for studying the epigenetic transgenerational effect in chickens, a suitable animal model. In order to effectively maximize the genetic gain of complex traits, breeding programs can now identify genes connected with thermotolerance as tolerant lines for heat stress. The fact that the majority of epigenomic studies on chickens have been conducted in closely supervised settings must also be considered in future studies. As a result, little is known about whether natural environmental modulation can cause adaptive epigenetic responses in chickens and whether such effects can last through thermally adapted progeny.

## Data Availability

The supporting findings of this study are available within the article.

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
