# Peer review of "Effects of Thermal Manipulation on mRNA Regulation of Response Genes Regarding Improvement of Thermotolerance Adaptation in Chickens during Embryogenesis"

_animals, 2022, doi:10.3390/ani12233354_

Round 1

Reviewer 1 Report

This review article focuses on comprehending and elucidating the cumulative effect of HS response genes, such as genes related to heat shock proteins, antioxidants, and immunological genes, which may contribute to the chickens' improved adaptability after being subjected to thermal manipulation during their embryonic stages.

This review is significant because it covers all relevant topics and offers current knowledge on potential molecular pathways that may aid in a better understanding of how thermal manipulation during chicken embryogenesis may enhance the development of thermotolerance. Every researcher in the field of thermal manipulation should take note of this review.

The conclusions offer a clear and current prospective molecular mechanism that may be in charge of enhancing the acquisition of thermotolerance.

The references are suitable for review subject.

The table needs to be formatted.

Please check the table 1 format.

Reviewer 2 Report

The main aim of the manuscript named “Thermal Manipulation on mRNA Regulation of Response Genes to Improve Thermotolerance Adaptation in Chicken during Embryogenesis” was to review the state of art about the cumulative effect of HS response genes, including different pathway (Hsp, antioxidant, and immune genes), which may contribute to the improved adaptability of chickens thermally manipulated in their embryonic stages. The present work is focus on the cumulative impact of HS in genes expression that explain by epigenetic changes. However, the review did not include specific works on epigenetics on this subject in chickens or in other species. This would be interesting to include this discussion to support the general hypothesis of the review.

Reviewer 3 Report

Dear Authors,

The topic is interesting and has relevance for the area. The aim and the scope of the review have been stated. I believe this MS would enhance our understanding and may indicate directions for the future research.

However following issues should be considered during the revision.

First of all, introduction part needs to be re-organized for fluency and repetitions should be prevented. Lines 66-69 have quite similar information to former paragraph (lines 64). These information better to be incorporated in the same paragraph. Then, gene expression deserves an independent paragraph.  

Line 67: ..egg development?  Please check if it is “egg” or “embryo”

Line 111: Title may include Hsp 60 as it has been a subheading later.

Line 127-128: Please check interpretation of Ali et al 2022. It seems TM induction days are wrong.  

Line 136-141: Please check interpretation of references 32 and 33.  There are contradictions with the results reported in these references.   

Line 301. 

Table 1: This table includes only 10 references. Therefore, it could be informative to explain if any methodological approach existed in rejection or acceptance of these references. Regarding to the main focus of this MS, it is understandable that there should be an attempt to keep a reasonable page numbers for introduction part.  However, by changing the table structure, the authors can incorporate the references reporting similar findings instead of giving results from each reference, separately.    Refernce #1: How TM affected embryo development (direction)?

Reference #3: EA and LA chicks exhibited what?

Reference #5: There is a mismatch between the reference cited and the information given in table for the reference (27).  Please check and revise accordingly. I think the information given in the table belongs to : Servet Yalçin, Sezen Özkan, Paul Siegel, Çigdem Yenisey, Mustafa Aksit, Manipulation of Incubation Temperatures to Increase Cold Resistance of Broilers:Influence on Embryo Development, Organ Weights, Hormones and Body Composition, The Journal of Poultry Science, 2012, Volume 49, Issue 2, Pages 133-139

Reference #6: ..lower body temperature

Reference #10: please check typos and rephrase for clarity
